# Experimental Comparison between Event and Global Shutter Cameras

**DOI:** 10.3390/s21041137

**Published:** 2021-02-06

**Authors:** Ondřej Holešovský, Radoslav Škoviera, Václav Hlaváč, Roman Vítek

**Affiliations:** 1Czech Institute of Informatics, Robotics and Cybernetics, Czech Technical University in Prague, Jugoslávských Partyzánů 1580/3, 160 00 Prague, Czech Republic; radoslav.skoviera@cvut.cz (R.Š.); vaclav.hlavac@cvut.cz (V.H.); 2Department of Weapons and Ammunition, Faculty of Military Technology, University of Defence, Kounicova 65, 662 10 Brno, Czech Republic; roman.vitek@unob.cz

**Keywords:** camera, event-camera, image sensors, pixel response

## Abstract

We compare event-cameras with fast (global shutter) frame-cameras experimentally, asking: “What is the application domain, in which an event-camera surpasses a fast frame-camera?” Surprisingly, finding the answer has been difficult. Our methodology was to test event- and frame-cameras on generic computer vision tasks where event-camera advantages should manifest. We used two methods: (1) a controlled, cheap, and easily reproducible experiment (observing a marker on a rotating disk at varying speeds); (2) selecting one challenging practical ballistic experiment (observing a flying bullet having a ground truth provided by an ultra-high-speed expensive frame-camera). The experimental results include sampling/detection rates and position estimation errors as functions of illuminance and motion speed; and the minimum pixel latency of two commercial state-of-the-art event-cameras (ATIS, DVS240). Event-cameras respond more slowly to positive than to negative large and sudden contrast changes. They outperformed a frame-camera in bandwidth efficiency in all our experiments. Both camera types provide comparable position estimation accuracy. The better event-camera was limited by pixel latency when tracking small objects, resulting in motion blur effects. Sensor bandwidth limited the event-camera in object recognition. However, future generations of event-cameras might alleviate bandwidth limitations.

## 1. Introduction and Task Formulation

This research extending our previous work [1] was motivated by our attempts to use an event-camera in robotics for interactive perception. Our practical experience with state-of-the-art event-cameras was inferior to that expected, and this work provides our explanation of why. Despite there being an abundance of publications reporting event-cameras in different use-cases, our contribution is novel in finding and understanding interesting phenomena when the event-camera works in limiting conditions. To that end, we designed an experiment involving markers on a rotating disk, which is easy to replicate. We suggest that cameras performing well in our experiments will also perform well in other computer vision applications, and that these experiments will be useful when evaluating future event-cameras.

Event-cameras, also known as Dynamic Vision Sensors (DVS), have been popular among academic researchers for about ten years. Independent pixels of event-cameras [2,3] generate asynchronous events in response to local logarithmic intensity changes. Each pixel performs level-crossing sampling of the difference of logarithmic brightness sensed by the pixel. Each time the difference passes a preset threshold, the pixel emits a change detection (CD) event and resets its brightness reference to the current brightness. A CD event is characterized by its pixel coordinates, its precise timestamp in microsecond resolution, and the polarity of the brightness change. The advantages of event cameras over traditional cameras include lower sensor latency, higher temporal resolution, higher dynamic range (120 dB+ vs. ∼60 dB of traditional cameras), implicit data compression, and lower power consumption.

Engineers or scientists wanting to use a vision sensor usually ask which kind of vision sensor is the most suitable for the task. When considering using event-cameras, one may ask questions such as *“What is the application domain, in which an event-camera surpasses a fast frame-camera?”* or *“Which scene conditions can be detrimental to event-camera performance?”*

Here, we contribute to answering these questions by experimentally comparing two different event-cameras with fast global shutter cameras. We selected marker recognition on a rotating disk and fast object tracking as use-cases, where we expected event-cameras to triumph. The experiments’ scope permits evaluation of the effects of sensor latency, temporal resolution, or bandwidth efficiency. To assess some of the event-camera performance limitations, we measure the latency of event response to different contrast stimuli at several sensor illuminance levels.

In terms of information coding efficiency, we found that an event-camera outperforms a frame-camera in all the tasks presented in this paper. Large and sudden positive contrast changes damage the performance of current event-cameras. Too low event readout bandwidth in the best-tested event-camera limits the range of speeds at which object recognition works well. Frame- and event-cameras perform similarly well in terms of the accuracy of moving object position estimation.

## 2. Related Work

We experimentally tested physical event- and frame-camera pixel arrays on scenes containing two distinct brightness levels and the transitions between them. We extended the rotating disk experiments started in [1] by adding accurate position ground truth and by recording the data for multiple distinct calibrated sensor illuminance levels. In addition to the marker recognition, the cameras tracked a simple object (a black dot) rotating on the disk. The analysis presented in this paper is extended in scope and detail. In our pixel latency experiments with two event-cameras, we present the latency of event response to negative and positive relative contrast steps across several contrast magnitudes and initial illuminance levels. Unlike the preliminary results in [1], the ballistic experiment in this work is also quantitatively evaluated and highly controlled.

The evaluation methodology is reported in [4], where high-speed CMOS/CCD frame-cameras are compared in a controlled experiment. The comparison methodology is similar to ours. They utilize six camera evaluation criteria: readout performance, camera sensitivity, intensity response linearity, image signal-to-noise ratio, electronic shutter performance, and image lag. We tested the first three of these, which are relevant to event-cameras.

Several authors deal with the comparison of event- and frame-cameras. Barrios-Avilés et al. [5] test the object detection latency of event and standard cameras. Their vision system detects a black circular dot rotating on a white disk and estimates the position of the dot for control purposes. Surprisingly, the authors report latency differences between the two cameras in the order of 100 ms, despite the frame rate of the standard camera being 64 fps at VGA image resolution. It is unlikely that such long latency would be caused by the cameras or by the object detection algorithm based on image intensity thresholding running on the standard camera frames.

Reconstruction of images from an event-camera is more complicated compared with frame-cameras. The state-of-the-art approach to cope with this task was published in Rebecq et al. [6], who compare the quality of images reconstructed from events to standard camera frames. The reconstructed images better capture the dynamic range of the scene than the standard frames. The authors also compare visual-inertial odometry algorithms running on traditional camera frames and images reconstructed from events. Event-based reconstructed intensity image results are reported to be on average superior to the results of traditional frames and the state-of-the-art methods running on events directly. However, the first is no surprise as the chosen traditional camera frame rate was only 20 frames per second, and the captured frames suffered from severe motion blur, probably due to overlong exposure time.

Boettiger [7] goes into great depth of analysis and comparison of the properties of both event- and frame-based cameras; the methodology relies on experimental evaluation similarly to ours. The author proposed a similar experiment of a rotating disk with a dot as used in our work. However, the frame-camera used sampled the scene with a relatively low frequency of 20 Hz. The analysis also did not include tracking objects moving at very high speed (e.g., our projectiles). Additionally, the experiments in this work were much less controlled (e.g., no control or precise ground truth for the rotational speed, no varying lighting conditions, etc.). The conclusion of the author, based on experiments, is that as of now, event-cameras are not significantly superior to frame-cameras in tracking application (at least not in general). The author concludes that further development of the event-cameras and tracking algorithms specific for asynchronous events is necessary.

Cox et al. [8] proposed to use Johnson’s criteria [9] to compare the automated target recognition performance of event- and frame-cameras. The authors modeled the bandwidth advantage of event-cameras over frame-cameras and assumed that performance is limited by sensor bandwidth. The theoretical analysis further presupposed that noise-free and highly sensitive event-cameras collect the same relevant information as the frame-cameras. We note that some of these assumptions may not be valid in practice, which motivates us to propose the experimental benchmark and analysis.

Censi et al. [10] proposed a power-performance approach to comparing sensor families on a given task. They applied the approach theoretically to the comparison of event-based and periodic sampling in a single-pixel vision sensor. The pixel latency and exposure time are neglected. The overall power consumption is a cost to be minimized and assumed to be linearly proportional to the available sensor bandwidth. The mean square brightness reconstruction error measured performance. The authors found that event-based sampling dominates periodic sampling across all power levels on brightness signals driven by a Brownian process and by sharp switches between two intensity levels (“Poisson” texture). On a very noisy, slowly varying signal, periodic sampling dominates. The two sampling methods perform equally well on a piece-wise linear signal (ramps). On a mixed piece-wise linear and “Poisson” texture signal, periodic sampling is better in the low-power regime, and event-based sampling is better when more power is available.

Event cameras can perform vibration measurement or monitoring [11,12]. Lai et al. [12] proposed to use a DAVIS240C event-camera in full-field structural monitoring, boundary condition identification and vibration analysis. The event-camera observed the free vibrations of a cantilever beam: the first natural frequency of the beam was approximately 10 Hz and the Photron Fastcam SA5 frame-based high-speed camera provided the ground truth data. The authors found that the advantages of the event-camera compared to the high-speed camera were the high dynamic range, the high equivalent frame rate and the absence of the blur effect.

We did not find any other comparison of frame- and event- cameras beyond the foregoing.

Articles introducing event-camera sensor designs usually test the sensors as well. Lichtsteiner et al. [3] measured the pixel transfer function and bandwidth as the mean event response to sinusoidal LED stimulation of 2:1 contrast across a range of frequencies for four DC levels of illumination. They also measured the latency of a pixel response to a positive 30% contrast step at a range of DC illuminance levels. Posch et al. [13] presented similar event latency measurements. They also evaluated pixel contrast sensitivity, which is the event probability due to increasing relative contrast at identical initial illuminance. Lichtsteiner et al. [3] estimated the standard deviation of the event contrast threshold, which is the only non-ideal behavior simulated by the open-source event-camera simulator ESIM [14].

Delbruck et al. [15] modelled contrast threshold uncertainty, pixel bandwidth, temporal noise, leak events, and hot pixels and used the model to convert videos to events. They noted that, due to the limited pixel bandwidth, a larger brightness step-change triggers a longer series of events, causing “motion blur”. Thus, it might not be possible to disambiguate the blur caused by motion from that caused by the finite response time under lower illumination conditions. Based on the pixel latency measured by [3], they discussed the DVS operation under natural lighting conditions. They recorded that negative (OFF) contrast edges cause lower latency than positive (ON) ones, but they did not quantify the difference. They did not model the finite event readout bandwidth.

Several high-speed applications using event cameras have been published. In [16], a pair of event cameras provided position feedback to keep a pencil balanced—the median rate of the feedback loop was 4 kHz. Delbruck et al. [17] built a robotic goalkeeper using event cameras, achieving a 550 Hz median update rate. Pacoret et al. [18] tracked microparticles at a frequency of several kHz by means of an event-based Hough transform. Howell et al. [19] applied event cameras to the detection and tracking of high-speed micrometer-sized particles in microfluidic devices.

The recent survey by Gallego et al. [20] mentions several different event-camera application areas such as real-time interaction systems, object tracking, surveillance, object recognition, depth estimation, optical flow, 3D structured light scanning, high dynamic range (HDR) imaging, video compression, visual odometry, and image deblurring. They also list properties of thirteen event-cameras. We used two of them in our experiments and one was the state-of-the-art commercially available camera. We discuss camera hardware related issues in more detail in the next section.

This paper proposes the first experimental, explicit, high-speed motion benchmark of event- and frame-cameras. Our experiments brought several novel findings: (a) The quantification of the ON/OFF response latency difference and its impact on high-speed object detection or tracking; (b) The implications of the burst-mode event readout [13,21] operating close to its bandwidth limit for the considered computer vision tasks; (c) The event-camera spatial sampling density monotonically decreases with growing motion speed. Therefore, fixed-sized event batches do not guarantee perfect speed-adaptive scene sampling as previously assumed [22], even if the scene appearance does not change; (d) The position estimation accuracy of frame and event-cameras is comparable when tracking small high-speed objects.

## 3. Materials and Methods

### 3.1. Methodologies, Experiments

We followed two experimental methodologies commonly used when comparing devices. The first explores a controlled experiment using a deliberately simple device, cheap, and easy to duplicate. We observed contrast patterns on a rotating disk at a controlled rotational speed. This approach differs from a rather expensive, high speed pulse lighting used for comparing high-speed frame-cameras in [4].

We printed a contrast black dot 1 mm (12 pixels) in diameter or five different 3×3 mm (36×36 pixels) ArUco markers (squares with a black background and a white generated 4×4 binary pattern [23], marker generation and detection software are available in OpenCV [24]), both on a white background, and pasted it to a rotating disk, at the radius of 85 mm. The disk was propelled by a BLDC servo motor D5065-270kv [25] controlled by an ODrive v3.6 controller [26]. An encoder CUI Devices (Lake Oswego, OR, USA) AMT102 [27] with 8192 counts per revolution tracked the motion of the motor shaft. An STMicroelectronics (Geneva, Switzerland) STM32 microcontroller unit (MCU) [28] performed the encoder readout and synchronization (see below for details). We could vary the number of revolutions in the range 0.5–40 rps, which corresponds to the dot/marker peripheral speed in the range 0.27–21 m/s and mean image speed 3.3–260 kpx/s. The scene was illuminated by an adjustable non-flickering panel light FOMEI (Prague, Czech Republic) LED WIFI-36D with color temperature set to 3700 K [29].

The second methodology tested event-cameras in a demanding practical use-case. We observed a flying bullet at high-speeds shot from a ballistic test barrel under controlled lighting, chosen because we found a collaborating ballistic laboratory specialized in testing personal firearms. This allowed us to test event-cameras at their speed limits. We measured related phenomena simultaneously with an expensive high-speed frame-camera. The speed of a bullet was measured independently by a Doppler radar along the bullet trajectory and by light gates at a distance of 2.5 m from the muzzle of the barrel. These two methods provided us the ground truth for the projectile speed.

### 3.2. Materials

We tested two event-cameras: iniVation (Zurich, Switzerland) DVS240 [30] (DVS240 in short), which is an evolved version of the popular DAVIS240 [31], and Prophesee (Paris, France) ATIS HVGA Gen3 [32] (ATIS in short). Posch et al. [13] presented an earlier generation of the ATIS sensor.

Table 1 in the event-camera survey [20] compares several commercial or prototype event-cameras. Some of them have better specifications than the two event-cameras of ours. However, we constructed our benchmark experiments such that only pixel and readout design affect event-camera performance. Larger camera pixel array resolution, for example, would not affect the reported performance metrics. Given our benchmark design, the DVS240 (DAVIS240) and DAVIS346 are still the best sensors produced by the company iniVation mentioned in the table. The Samsung (Seoul, South Korea) cameras were not commercially available in 2020: the exception we found was the “Samsung SmartThings Vision” home monitoring device, which has an event-camera embedded inside. However, we did not find a simple way of connecting the embedded camera to a computer and recording the events it emits. Before buying the Prophesee ATIS camera, we briefly experimented with the CelePixel (Shanghai, China) CeleX-IV camera. Although its specifications on paper are impressive, it performed much worse in our initial test than the first iniVation product commercially available, the DVS128 from 2008. Prophesee told us in May 2020 that they planned to release their Gen 4 CD sensor in Q4 2020 or later, and so we could not test it. These findings make us believe that the Prophesee Gen3 ATIS remains one of the state-of-the-art commercially available event-cameras as of 2020.

The cameras we tested have been widely used by researchers and so are relevant to a large scientific community. Event-camera users may use our benchmark to test newer cameras.

We observed the phenomena in the rotating disk experiments with a global shutter frame-camera Basler (Ahrensburg, Germany) ACE acA640-750um [33] (Basler in short). In the shooting experiment, we used the global shutter frame-camera Photron (Tokyo, Japan) Fastcam SA-Z [34] (Photron in short).

We used a ballistic Doppler radar Prototypa (Brno, Czech Republic) DRS-01 [35] together with Kistler (Winterthur, Switzerland) Type 2521A [36] light gates for independent measurement of the bullet speed. Furthermore, the light gates also provided the trigger signal for the synchronization of the camera records. Because the output signal from the light gates is of an irregular shape, we used the programmable triggering unit Prototypa PTU-01 [37] to create a regular rectangle impulse based on the light gate output signal. The rising edge of this rectangular impulse triggered the record of the tested cameras. The Veritaslight (Pasadena, California) Constellation 120 [38] LED lights and DedoLight (Munich, Germany) Dedocool [39] tungsten lights illuminated the scene during the ballistic measurements.

The rotating disk experiment is illustrated in Figure 1. Figure 2 depicts the ballistic experimental setup schematically. Figure 3 shows the corresponding photo of the laboratory shooting range.

Table 1 summarizes the basic parameters of the cameras used in our experiments. The stated Basler exposure time and frame rate are the fastest possible settings used at the highest tested sensor illuminance of 1000 lx. Increasing the illuminance to 2 klx resulted in overexposure. At weaker illuminance levels, the Basler exposure time was 1270 µs at 20 lx, 274 µs at 80 lx, and 59 µs at 400 lx, yielding a mean pixel brightness value of 42.0 on the white paper surface.

It is important to compare the cameras on the same scenes with the same scale and sensor illuminance to avoid misleading conclusions. We observed these recommendations in all experiments with one minor exception in the ballistic experiment. The Photron camera had weaker sensor illuminance than the event-cameras.

Ideally, the pixel photodiode area in all the tested cameras should be the same. Unfortunately, it is not easy to find multiple different event- and frame-cameras with the same photodiode size.

### 3.3. Illuminance Measurement

We used a Basler Dart daA2500-14um frame-camera [40] (Basler Dart in short) in tandem with a Sekonic (Tokyo, Japan) Speedmaster L-858D light meter [41] (Sekonic light meter) for measuring the sensor chip illuminance in the tested cameras.

The exposure time the Basler Dart camera needs to obtain the mean brightness value of 127 (half of the brightness range) is inversely proportional to the sensor illuminance. We calibrated the exposure times to the standard illuminance scale (in lux) using the Sekonic light meter. During calibration, both the lens-free Basler Dart camera and the light meter were pointed at an illuminated white wall from the same distance.

### 3.4. Event Pixel Response Measurement

In order to separate the event pixel response limitations from the sensor readout limitations as much as possible, we performed additional event pixel response measurements.

A pre-programmed arbitrary waveform generator controlling an LED produced steps in relative contrast as the stimuli for a tested event-camera.

To verify the resulting LED contrast levels, we mounted a photodiode opposite the LED. Using an oscilloscope, we measured the photocurrent flowing through the photodiode, which is proportional to the illuminance.

The waveform generator was synchronized with the event-camera at each stimulus onset using external trigger signals sent to the event-camera.

We tested the camera in a completely dark room without a lens in this experiment to ensure homogeneous and fully controlled illuminance across all its pixels.

To prevent event readout congestion, we activated only a small rectangular pixel region of interest (ROI) in the event-camera chip. However, to make the experimental results more externally valid, the region of interest should not be too small. Electronic noise or event readout interactions among neighboring pixels may affect the observed pixel response in a real-world scenario. We chose a 20–40 pixel large ROI so that one event may be read from each pixel in 2–4 µs at a 10 Meps event rate.

An event-camera might assign the same timestamp to events emitted within a short time period. To avoid this potential pitfall, we recorded the event-camera response to contrast stimuli of different duration (from 2 µs up to 3 ms). This way, only sufficiently fast events emitted during the stimulus presence contributed to the measured pixel response.

### 3.5. Rotating Disk

The rotating disk allows us to test high-speed visual sensing with a rotary encoder’s accurate position ground truth. Storing the camera measurements for all rotary encoder positions at low speed gives the position ground truth for the camera measurements at higher speeds.

Data recording in general works as follows. The above-mentioned STM32 microcontroller unit (MCU) reads the encoder position at the beginning of every global shutter exposure. In the event-camera case, the MCU reads the encoder regularly (at 16 kHz at most). When the encoder is read, the MCU sends an external trigger signal to the event-camera. The event-camera captures the timestamp of the trigger signal and sends it over the USB interface together with the DVS events timestamped by the same clock.

### 3.6. Intensity Reconstruction and Marker Detection

To test all the cameras on a simple pattern recognition task, we chose to detect ArUco markers [23] rotating on the disk.

In the case of the event-cameras, the markers are detected in intensity images reconstructed from events. We used a state-of-the-art intensity reconstruction method called E2VID described in [6,42]. Code is available [43].

The E2VID method uses a recurrent convolutional neural network whose architecture is similar to UNet. In each iteration, the network computes a reconstructed intensity image as a function of a batch of events and a sequence of several previously reconstructed intensity images. Rebecq et al. stored each event batch for the network input into a spatio-temporal voxel grid. The network was trained in a supervised mode on simulated event sequences and corresponding ground-truth intensity images.

A faster and smaller neural network version of E2VID was published in [44]. More recently, Ref. [45] outperformed E2VID on certain event datasets by training the neural network on augmented simulated data with a wider range of event rates and contrast thresholds. We did not use [45] as we found it after we had processed our experimental data using E2VID.

### 3.7. Dot Position Estimation

We need a simple but robust method to estimate the rotating dot position at wide ranges of event noise, image speed, and illuminance.

To obtain speed-invariant scene edges in a constant appearance scene, we accumulate a constant number of events for each position estimate [22]. The accumulated events should at least cover the entire dot circumference to enable reliable position estimation.

Our pixel response experimental results suggest that event-cameras respond more quickly to negative than to positive contrasts. Thus, we accumulated only the negative events in the rotating dot and ballistic experiments. Their median coordinates yielded the dot position estimate.

In the case of the Basler frame-camera, we selected the black dot pixels using global thresholding. The black dot median coordinates yielded the position estimate. The event-camera position estimates may be offset from the frame-camera estimates. This does not matter in the rotating disk experiment, where each camera generates its own position ground truth at low speeds. The offsets do not affect projectile speed estimation in the ballistic experiment. The offsets only influence the position estimation, which we discuss together with the ballistic experimental results.

### 3.8. Ballistic Experiment

We conducted the ballistic experiment to estimate the potential of event-cameras in tracking and speed measurement of a high-speed object, in our case, the shot projectile.

We used the following cartridges in the experiments: (a) a 9 × 19 mm pistol cartridge, nominal speed $v_{0}=360$ ms${}^{-1}$; (b) a 9 × 19 mm pistol cartridge, reduced speed $v_{0}\;\approx \;100$ ms${}^{-1}$; (c) a 7.62 × 51 mm rifle cartridge, nominal speed $v_{0}=850$ ms${}^{-1}$.

The cartridges were shot from ballistic test barrels. The projectile velocity has been measured using a Doppler radar and the light gates as a reference.

We compensated for lens distortion and calibrated the cameras in a single world coordinate frame using a chessboard calibration pattern to obtain comparable position estimates from all the cameras. We synchronized the cameras using the pulse signal from a projectile-detecting light gate between the gun-barrel and the camera field of view. Each projectile velocity estimate was calculated as the numerical differentiation of the projectile position.

In the frame-camera case, the projectile image coordinates were determined by global thresholding.

Accumulating a constant number of events for each position estimate resulted in significantly higher dispersion of the instantaneous velocity estimates than accumulating events for a constant time. Therefore, we estimated the projectile image coordinates as the centroid of the negative polarity events accumulated in a constant temporal interval.

## 4. Results

We present the minimum measured latency across illuminance and contrast of both tested event-cameras in this section. The rotating disk results compare the sampling/detection rates and densities of the tested cameras as functions of image speed and illuminance. We report position estimation errors in the rotating dot experiment. The ballistic results include speed and position estimation errors. We support these quantitative results qualitatively by showing sample images. Finally, we compare the data efficiency of event- and frame-cameras on the rotating dot and rotating marker tasks.

### 4.1. Pixel Latency in Event-Cameras

The minimum pixel latency is the shortest stimulus duration required for an average pixel to emit one event with probability at least ≥0.5.

Larger relative contrast magnitude and larger illuminance cause lower minimum latency, see Figure 4. We observed that the latency of negative polarity events is lower than the latency of positive polarity events given the same absolute contrast. The ATIS sensor consistently outperformed the DVS240 in terms of the minimum pixel latency.

The shortest latency was 4 µs in the ATIS and 100 µs in the DVS240 sensor, both at the −100% contrast and the strongest tested reference illuminance of 2000 lx. Our ATIS result almost matches the best latency of 3 µs reported in [13]. However, we were not able to reproduce the 3 µs latency reported in [31], which describes the DAVIS240 sensor, the preceding version of the DVS240 camera tested by us.

### 4.2. Rotating Dot Experiment

The black dot has a diameter equal to 12 pixels in our experimental setup, cf. Figure 1. We assume that 19 events are needed at least to cover the half of the dot circumference. We chose to accumulate $N_{c}=100$ negative polarity events to increase robustness to noise events or event readout failures.

The rotating dot position estimate error grows with increasing image speed and decreasing illuminance in general. Figure 5 shows the best results achieved for each camera and illuminance level.

However, we noticed several exceptions to the general rule for the DVS240 camera. Surprisingly, the position error is approximately twice as large with the 2000 lx illuminance as with the 400 lx for image speeds below 40 kilo-pixels per second (kpx/s). Furthermore, there seems to be a short speed interval around 80 kpx/s where the position estimation error plot lines of all the illuminance levels intersect.

The sharp error peaks at around 150 kpx/s in the ATIS and Basler subplot of Figure 5 were caused by mechanical resonance of the rotating disk at the respective rotational velocity of ca. 22 revolutions per second.

The dot position estimation error with the Basler camera at the strongest illuminance is comparable to the ATIS result, whereas it is worse at weaker illuminance (Figure 5). However, we cannot conclude that event cameras are inherently better than global shutter cameras at weaker illuminance because the photodiode area is at least four times smaller in the Basler than in the ATIS pixels, see Table 1.

We needed to set a higher ATIS pixel sensitivity s=50 at lower illuminance levels (20 and 80 lx) than at higher illuminance levels (400 and 2000 lx, sensitivity s=40) in order to obtain the lowest position errors presented. In contrast, we obtained the lowest DVS240 errors with the same bias setting.

With the ATIS sensor, events spread more in space due to increasing speed or decreasing illuminance level, see sample ATIS images in Figure 6. The prolonged and more uncertain edges of the event-images resemble the effects of motion blur in images from the Basler global shutter camera in Figure 7. The positive polarity edges tend to be blurred more than the negative ones, especially at higher speeds and lower illuminance levels.

All the DVS240 samples in Figure 6 contain a long tail of positive polarity events. Higher speeds often cause captured events to spread across a couple of pixel rows. At speeds around 200 kpx/s, a distinct but blurred group of negative polarity events was seen only at the highest illuminance level.

As the position estimation method is event-based, the effective temporal sampling rate automatically increases with image speed in both event-cameras, see Figure 8. The Basler global shutter sampling rate remains fixed but adapted to the exposure time of each illuminance level. The lower sensitivity setting of 40 in the ATIS causes a lower sampling rate, as it takes more time to collect a constant number of events with lower sensitivity than with a higher one. Note the general trend of higher sampling rates at weaker illuminance levels in both cameras. Finally, when the pixels gradually stop detecting contrast at high speeds, the sampling rate decays.

Although the event-camera temporal sampling rate increases in general with image speed, the increase is too slow to preserve a constant spatial sampling density. The spatial sampling density monotonically decreases with increasing speed, see Figure 9 showing the number of independent samples obtained per 100 pixels of distance travelled by the rotating disk surface. This may be related to the observation of the area occupied by a fixed number of events increasing with increasing speed in the images of Figure 6.

### 4.3. Rotating Marker Experiment

To maximize marker detection density, we need to optimize the event batch size in the E2VID method used for reconstructing the intensity frames. The smaller the batch size, the more markers per disk revolution can be detected at lower speeds. However, larger batches provide for a more reliable intensity reconstruction at higher speeds. Of the batch sizes tested (270, 540, 1080, 2160, 4320 and 8640), we chose a size of 2160 events for the ATIS and 540 events for the 20 and 80 lx DVS240 recordings. These sizes enable marker detection at the highest speeds while maximizing detection density at lower speeds. We had to use 1080-sized event batches to achieve reliable low-speed marker detection in the 400 and 2000 lx DVS240 recordings.

The rotating marker results mostly resemble the rotating dot results. Spatial marker detection density in Figure 10 and the spatial dot sampling density in Figure 9 both decay with growing speed.

Likewise, the detection frequency of the event-cameras in Figure 11 grows with growing speed until reaching a maximum. Further increase in speed causes lower detection frequency. Compare with Figure 8.

We note that illuminance’s impact on the ATIS detection performance is much lower than in the rotating dot experiment. This suggests that different factors limit the ATIS performance in each task.

The sample images in Figure 12 and Figure 13 confirm these observations. The DVS240 image quality degrades gradually across the speed range. In contrast, the ATIS samples are almost perfect for most of the speed range, degrading sharply around 40 kpx/s. Furthermore, the ATIS image quality is less affected by illuminance than the one of the DVS240.

Marker detection is possible in (almost) every frame captured by the global shutter Basler camera, yielding a constant detection frequency equal to the frame rate, see Figure 11. Decreasing sensor illuminance demands longer exposure time, which causes stronger motion blur at a given speed, eventually resulting in marker detection failure. Furthermore, the exposure time restricts from below the interval between subsequent frames, limiting the frame rate at weaker illuminance.

Both event-cameras outperform the 1000 Hz Basler camera in the maximum achieved detection frequency. However, the Basler camera keeps detecting some markers at approximately four times higher speeds than the best event camera tested (ATIS).

Figure 7 shows sample Basler dot and marker images at the strongest illuminance level for three speeds. The stronger motion blur at higher speeds is clearly visible. For example, the marker moving at 209 kpx/s translated by approximately twelve pixels (1/3 of the length of its side) during the 59 µs exposure time. Nevertheless, it can still sometimes be detected and read by the detection algorithm.

### 4.4. Ballistic Experiment

The ATIS and DVS240 sensor illuminance was ca. 4500 lx in the ballistic experiment. Table 2 details the metric and image dimensions of the two projectile types we used. Sample images recorded by the Photron, ATIS, and DVS240 cameras at three projectile speeds are shown in Figure 14. The projectiles flew from the right to the left. While the effects of increasing speed are not visible in the 1 µs exposure images from the Photron camera, the leading negative polarity edges in the ATIS images become more imprecise with increasing projectile speed. In addition, the trailing positive polarity events extend over more pixels at a higher speed. In our study, the DVS240 camera could not capture the projectile’s position or appearance even at the lowest tested speed, where the 10 µs long event window contained events spread across a single pixel row.

The projectile position estimates from the ATIS event camera are very close to the ground truth Photron estimates, see Figure 15a. In the 365 m/s or slower recordings, the relative position variability is below 0.7% of the trajectory length. This is 3 mm or 2.5 px, and the distance traveled by the projectile in 8 µs. In the slowest 100 m/s shot, the events are accumulated mostly on the projectile’s frontal edge during the 10 µs interval; see the sample ATIS event image in Figure 14. The thinner edge probably causes the systematic position shift ahead of the Photron estimates. In the 850 m/s experiment, the ATIS position estimates systematically lag behind the Photron estimates by ca. five millimeters.

The other contribution to the systematic shift between the projectile positions estimated from the Photron and the ATIS record could be the difference between the calibration plane and the real plane of the projectile motion. This difference is caused by the projectile trajectory dispersion.

Because numerical differentiation is very sensitive to noise, the relatively low noise in the determination of the projectile position results in large uncertainty of the estimated immediate projectile velocity, as shown in Figure 15b and Figure 16. The ATIS immediate speed estimates exhibit significantly larger uncertainty at all the three tested projectile velocities than the Photron estimates.

However, Table 3 shows that the mean projectile velocity values determined using the event-camera and the Photron camera records agree. The uncertainty reported for both cameras in each of the three shots is the standard deviation of the immediate speed estimates from the mean speed. The highest relative difference between the measurements was 1.6% in the 850 m/s shot. Furthermore, both camera estimates of the mean projectile velocity correspond very well with the ballistic Doppler radar measurements.

We observe that, to our knowledge, these reports of using an event-camera in practical ballistic measurements are the first.

### 4.5. Data Efficiency

We compared the data efficiency of the ATIS event-camera and an idealized frame-camera of the same resolution in Figure 17. The information of interest, a detected marker or a dot position, is produced at the output sampling rate. The output rate is a function of sensor bandwidth, i.e., the amount of data per second sent by a sensor to the computer for processing. This function is task-independent for the frame-camera as each frame yields one output sample in the tasks we assumed.

We supposed that one event can be stored in approximately 27 bits. To encode the pixel coordinates, one needs $\left\lceil \log_{2}\left(N_{pixels}\right)\right\rceil =\left\lceil \log_{2}\left(480\times 360\right)\right\rceil =18$ bits. 1 bit is required to store the polarity and 8 bits for a 0–255 µs timestamp. (To keep track of time in longer recordings, a 32 bit long timestamp prefix can be sent every 256 µs, for example, increasing the sensor bandwidth only by 125 kbits/s.) If the entire camera and computer system are neuromorphic, there is no need to assign timestamps to the events [2].

Under these conditions, the ATIS event-camera was more data-efficient than the frame-camera, by two orders of magnitude in the rotating dot task and by one order of magnitude in the marker detection task. The ATIS marker detection curve in Figure 17 starts to decrease around 270 Mbits/s or 10 million events per second, probably due to hardware bandwidth constraints of the ATIS sensor.

## 5. Discussion

Compared to an event-camera, a global shutter frame-camera could be more efficient in more complex scenes than in the simple scenes we tested. In the following illustration, we assume the sensor pixel resolution to be at most 512×512 pixels, requiring 27 bits per event on average and eight bits per one intensity sample. A larger resolution favors the frame-camera, as the events need more bits to encode the pixel coordinates. Once the number of instantaneously sampled pixels required to produce one output sample exceeds 8/27=30% of the entire pixel array, the frame-based data encoding scheme becomes more efficient. When all pixels need to emit one event, e.g., when the global illuminance changes, the global shutter will be 3.4× more efficient. Additionally, when multiple events per pixel need to be captured to recognize multiple contrast levels or large and sudden illuminance changes, frame-based sampling will be even more suitable.

Event readout aggregated by pixel rows (“burst-mode” arbiter [13,21]) trades lower bandwidth for lower timestamp accuracy when many pixels in the same row emit events approximately at the same time. In that case, the sensor sends the pixel column coordinate and the event polarity of each event. The common row coordinate and the timestamp need to be sent only once per row.

We observed the readout aggregation in action at large image speeds when the sensor readout capacity limits were reached. Both the ATIS and DVS240 event-cameras read multiple events from a single pixel row at the same time. See, for example, the fastest dots and the slowest ballistic projectile recorded by the DVS240 in Figure 6 and Figure 14 or the fastest markers recorded by the ATIS in Figure 12. In [1], we observed that these readout limitations prevent accurate tracking of ballistic projectiles flying along pixel columns instead of pixel rows of the ATIS camera.

The event-based readout can be both a blessing and a curse. On the one hand, it adapts the event camera to the scene dynamics. On the other hand, the camera cannot collect a brief, complete, on-demand snapshot of a rapidly changing large scene. Future event-cameras with extended readout bandwidth and sufficiently large on-chip memory buffers could alleviate this limitation.

Position estimation error tends to grow with increasing speed in the rotating dot experiment, even though a constant number of events is accumulated to obtain each estimate. This suggests that perhaps the error could be reduced in the event-camera records by explicit edge sharpening such as event lifetime estimation [46,47]. At lower speeds and stronger illuminance levels (up to ca. 40 kpx/s at 2000 lx), the ATIS negative event timestamps consistently increased from the leading to the trailing edge of the dot, suggesting that edge sharpening could be applied. However, only the estimation accuracy at the higher speeds and/or lower illuminance scenarios could significantly benefit from the edge sharpening. Unfortunately, we did not notice a systematic pattern in the event timestamps recorded at the higher speed and/or lower illuminance scenarios. We interpret this observation as event-camera “motion blur”.

The event-camera was slow at detecting large positive relative contrasts, i.e., transitions from dark to bright illuminance, for three reasons. First, such contrasts can be almost infinite, demanding an infinite event response. Second, the minimum event pixel latency is longer at lower illuminance levels, prolonging the total event response duration. Third, in general, the minimum pixel latency is longer for the positive than for the negative contrast stimuli.

As expected, keeping the scene appearance fixed, the fixed-size event packet measurement mode increased the output sampling rate with increasing stimulus speed. At the same time, however, the spatial sampling density monotonically decreased with increasing speed. This implies fixed-sized event batches surprisingly do not guarantee perfect speed-adaptive scene sampling, as assumed, for example, by [22].

The dot position estimation error increases with growing image speed in all the cameras tested. This may be related to the spatial sampling density decreasing with the growing speed in all the cameras.

The ATIS event camera measured a ballistic projectile’s mean velocity at an accuracy comparable to the very high-speed 100 kHz global shutter Photron camera. However, the ATIS was significantly worse than the Photron at measuring the instantaneous projectile velocity.

Our ballistic experiment validated the rotating disk experiment results. These experiments do not differ qualitatively. We performed the ballistic experiment to test quantitatively higher image speed, which is 730 kpx/s compared to at most 300 kpx/s in the rotating disk experiment.

## 6. Conclusions

Event-cameras’ performance was limited by pixel latency when tracking small objects and by readout bandwidth in object recognition. When comparing the event- to frame-cameras, we saw analogies between event pixel latency and exposure time and event readout bandwidth and frame rate.

We provide an answer to the question “What is the application domain, in which an event-camera surpasses a fast frame-camera?”. In this study, the event-camera surpassed the fast frame-camera in terms of information coding efficiency in scenes with significant changes restricted to less than 30% of the field of view within the sampling period of interest. As the event pixel latency is significantly lower for negative than for positive contrast changes, the fastest scene changes should ideally be restricted to the negative contrast.

On the other hand, the coding efficiency analysis suggests that a frame-camera may outperform event-cameras in cluttered scenes. Assuming the readout bandwidth is fixed, the event-camera output sampling rate decreases as the scene complexity/clutter increases, whereas the frame-camera output sampling rate remains constant. We answer the question “Which scene conditions can be detrimental to event-camera performance?”: highly cluttered scenes or scenes with sharp and strong positive contrast edges can be detrimental to event-camera performance.

Although a frame-camera is more limited by exposure time than bandwidth in high-speed object recognition tasks, the better event-camera we tested (ATIS) was limited more by its readout bandwidth. Future event-cameras with extended readout bandwidth and sufficiently large on-chip buffers could alleviate this limitation. Nevertheless, the potential and future perspective of the event-based principle of sensing is obvious and is emphasized, e.g., by a related DARPA Fast Event-based Neuromorphic Camera and Electronics (FENCE) program from December 2020 [48].

Surprisingly, the event-camera spatial sampling density monotonically decreases with growing motion speed. This fact may limit the applicability of existing event-based algorithms relying on fixed-sized event batches.

Finally, our results showed that frame- and event-cameras perform similarly well in moving object position estimation accuracy.

In future work, (a) we aim to research event/frame-camera performance in high dynamic range scenes; and (b) use event-cameras in robotic perception tasks requiring fast feedback loops.

## Figures and Tables

**Figure 1 sensors-21-01137-f001:**
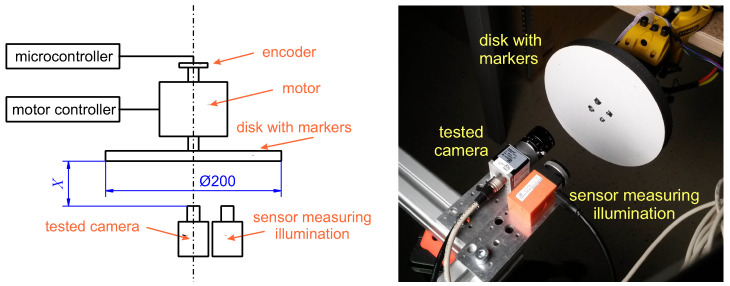
(**Left**) Schematic layout of the rotating disk experiment. The panel light is behind the cameras; (**Right**) a photograph of the rotating disk experiment.

**Figure 2 sensors-21-01137-f002:**
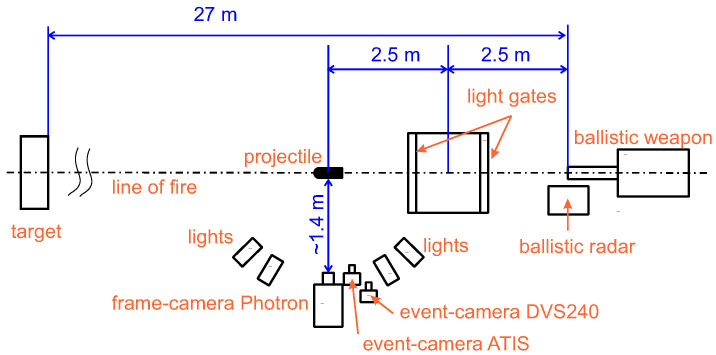
Schematic layout of the ballistic experiment.

**Figure 3 sensors-21-01137-f003:**
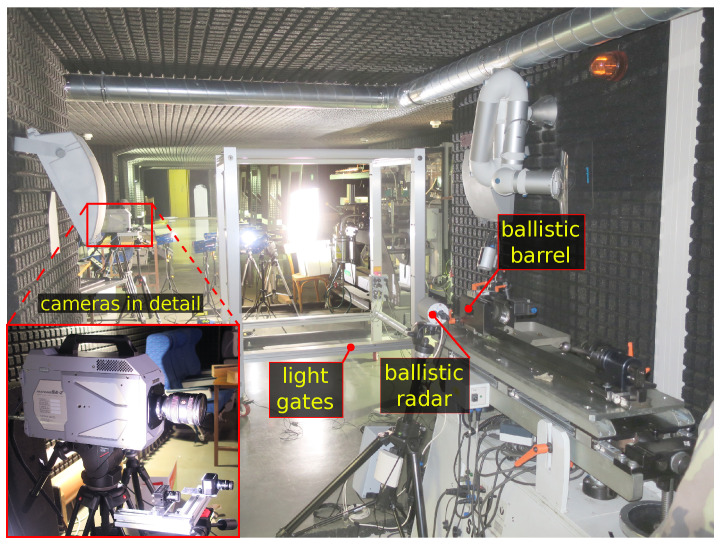
The ballistic shooting range setup.

**Figure 4 sensors-21-01137-f004:**
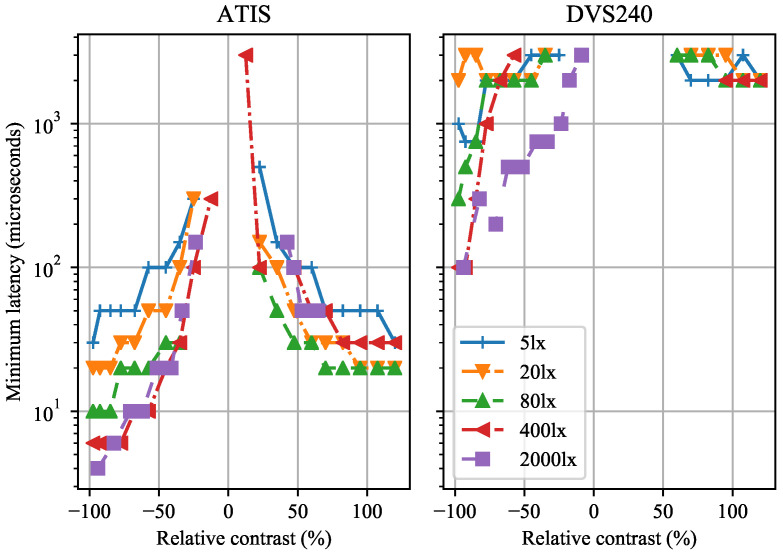
The minimum pixel latency of the ATIS and DVS240 event cameras depends on the contrast and illuminance (see the legend).

**Figure 5 sensors-21-01137-f005:**
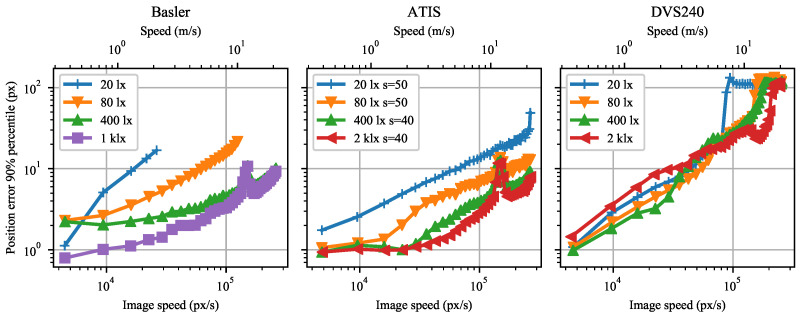
90% percentile of the position estimation error as a function of dot image speed for Basler, ATIS, and DVS240 cameras. Data were measured for four background scene illuminance levels at the sensor plane.

**Figure 6 sensors-21-01137-f006:**
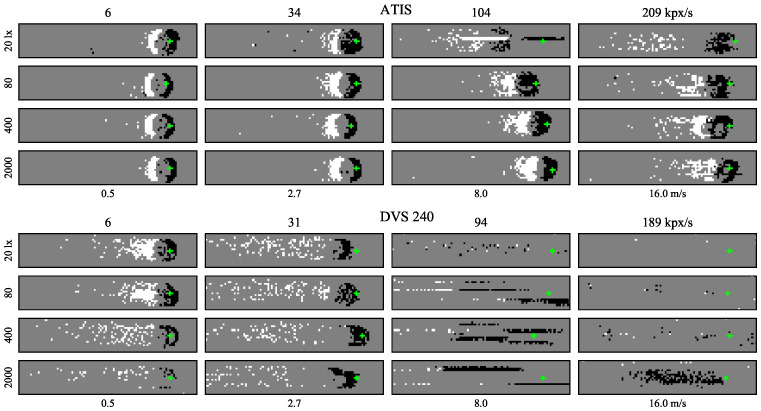
Sample ATIS (**top**) and DVS240 (**bottom**) event images of the black rotating dot moving left to right on white background at four different speeds (horizontal axis) and four different sensor illuminance levels (vertical axis). All images are cropped to 95×15 pixels from the original sensor plane with 200 events. Positive polarity events are white, negative black, and the background is grey. The green crosses indicate the ground truth median coordinates of the negative polarity events.

**Figure 7 sensors-21-01137-f007:**
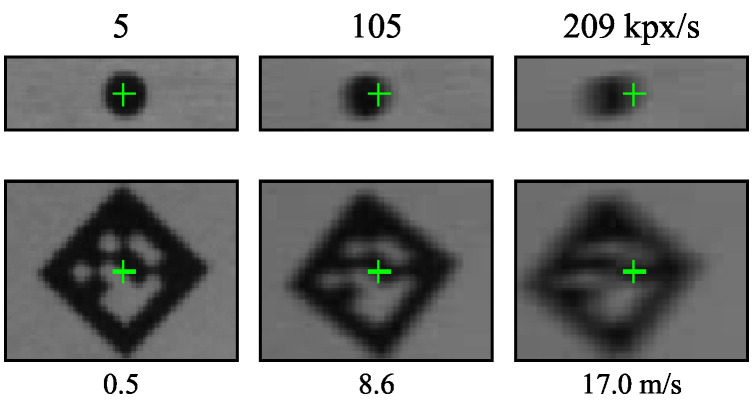
Sample images from the global shutter Basler camera. The green crosses show the ground truth position of the dot or marker center at the start of exposure. Exposure time 59 µs, sensor illuminance 1000 lx.

**Figure 8 sensors-21-01137-f008:**
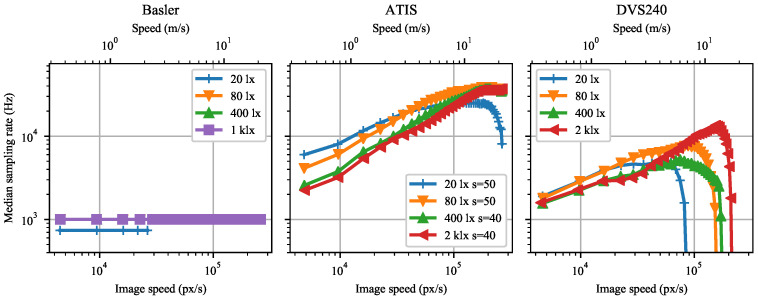
The effective median position sampling rate as a function of dot image speed for Basler, ATIS, and DVS240 cameras when computing each position estimate from 100 negative polarity events. Data were measured for four background scene illuminance levels at the sensor plane.

**Figure 9 sensors-21-01137-f009:**
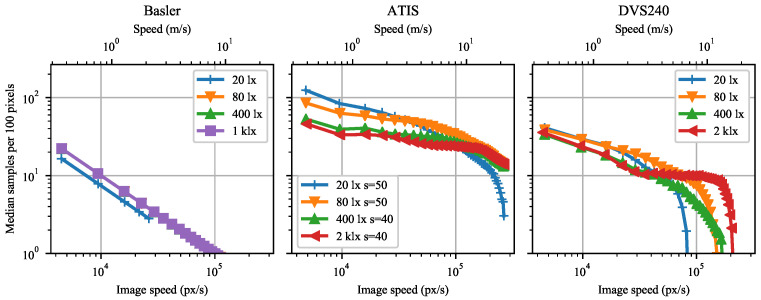
The median spatial sampling density in position estimates per 100 pixels distance as a function of dot image speed for Basler, ATIS, and DVS240 cameras. Each position estimate is computed from 100 negative polarity events. Data were measured for four background scene illuminance levels at the sensor plane.

**Figure 10 sensors-21-01137-f010:**
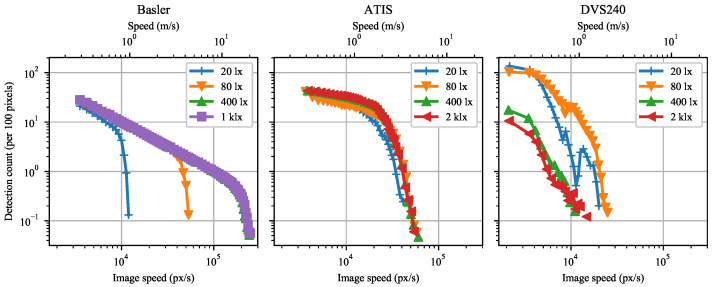
Mean number of marker detections per 100 pixels of marker trajectory as a function of marker speed for Basler, ATIS, and DVS240 cameras. Data were measured for four background scene illuminance levels at the sensor plane.

**Figure 11 sensors-21-01137-f011:**
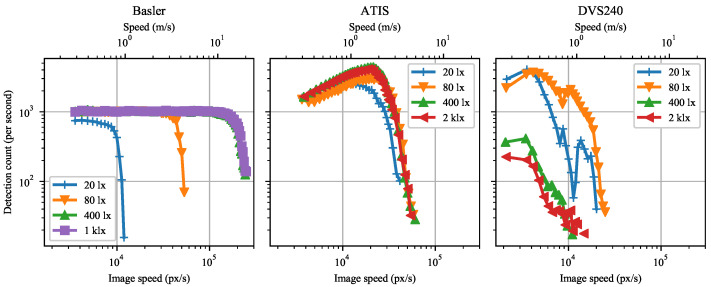
Mean number of marker detections per second as a function of marker speed for Basler, ATIS, and DVS240 cameras. Data were measured for four background scene illuminance levels at the sensor plane.

**Figure 12 sensors-21-01137-f012:**
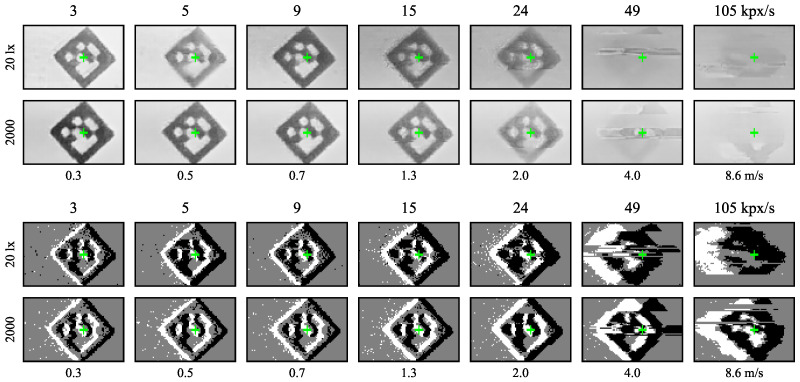
Sample reconstructed intensity and event marker images from the ATIS sensor. The images are shown for two illuminance levels at the sensor plane and seven image (**top**)/metric (**bottom**) speeds. The green crosses indicate the ground truth position of the marker center.

**Figure 13 sensors-21-01137-f013:**
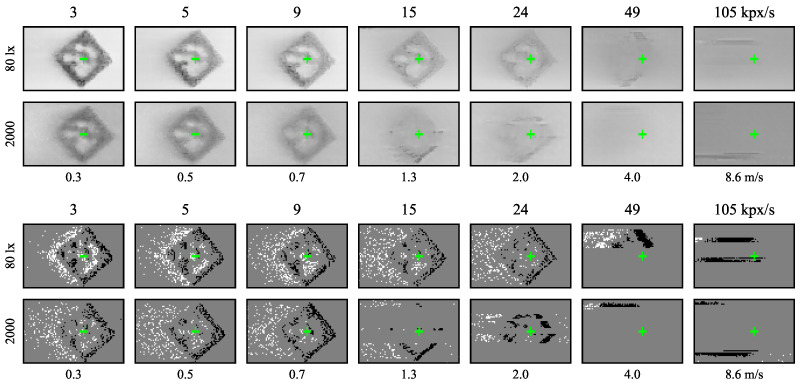
Sample reconstructed intensity and event marker images from the DVS240 sensor. The images are shown for two illuminance levels at the sensor plane and seven image (**top**)/metric (**bottom**) speeds. The green crosses indicate the ground truth position of the marker center.

**Figure 14 sensors-21-01137-f014:**
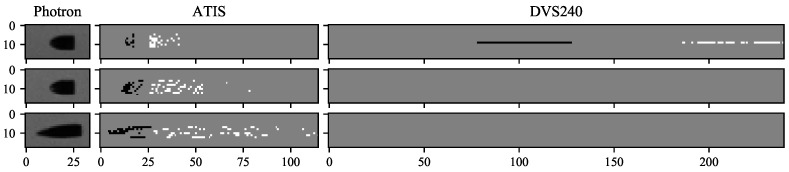
Sample images of the projectiles as seen by the Photron, ATIS, and DVS240 cameras. The event camera images display 10 µs of events, white are positive and black are negative polarity. The Photron exposure time was 1 µs. The projectile speed increased from top to bottom, from 100 through 365 to 850 m/s. (Image speed 87, 310 and 730 kpx/s.)

**Figure 15 sensors-21-01137-f015:**
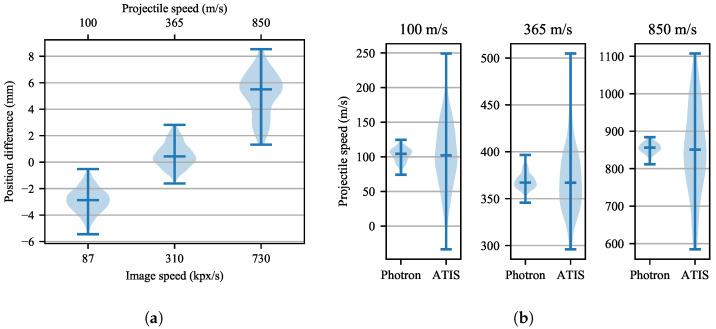
(**a**) Distributions of the differences between the ATIS and Photron estimates of horizontal projectile position. Data were measured for three projectile speeds along a trajectory segment 440 mm long. A positive difference means that the ATIS estimate lags in time behind the Photron estimate. The horizontal lines of each violin show the maximum, median, and minimum position differences; (**b**) distributions of the ATIS and Photron projectile speed estimates along a trajectory segment 440 mm long.

**Figure 16 sensors-21-01137-f016:**
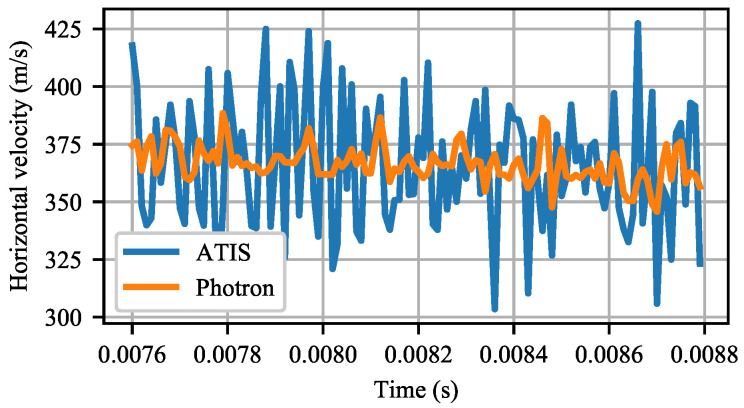
Horizontal projectile velocity estimated by the Photron and ATIS cameras along a 440 mm long trajectory segment at a constant 100 kHz sampling rate.

**Figure 17 sensors-21-01137-f017:**
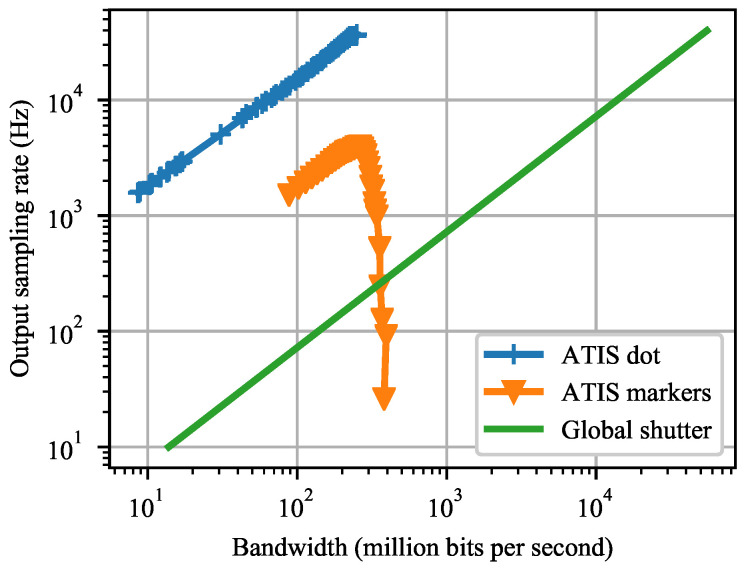
A bandwidth-performance plot shows how much data an event-camera (ATIS) and a global shutter frame-camera generate to reach a certain output sampling rate. The output sampling rate captures the number of marker detections or the number of rotating dot position estimates, both per second—assuming the strongest lighting tested (ATIS), 27 bits per event on average, negligible frame-camera exposure time, and sensor resolution of 480×360 pixels.

**Table 1 sensors-21-01137-t001:** Camera parameters. Meps—million events per second, FPS—frames per second.

Camera	DVS240	ATIS	Photron	Basler
resolution [px]	240×180	480×360	640×280	480×360
pixel size [µm]	18.5×18.5	20×20	20×20	4.8×4.8
fill factor [%]	22	25	58	Unknown
exposure [µs]	N/A	N/A	1	59
frame rate [FPS]	N/A	N/A	100,000	1000
max. event rate [Meps]	12	25	N/A	N/A

**Table 2 sensors-21-01137-t002:** Dimensions of projectiles in the world and image units.

Projectile	World Diameter [mm]	World Length [mm]	Image Diameter PHOTRON [pixel]	Image Length PHOTRON [pixel]	Image Diameter ATIS [pixel]	Image Length ATIS [pixel]
9 mm FMJ	9.03	15.8	7.50	13.14	7.92	13.83
7.62 mm M80	7.83	29.46	6.50	24.50	6.89	25.79

**Table 3 sensors-21-01137-t003:** Mean horizontal projectile speed estimates.

Mean Horizontal Speed (Meters per Second)		
**Radar**	**Photron**	**ATIS**
102.8	103.8±9.2	103.0±47.2
364.5	368.4±7.0	363.7±30.2
850.5	859.6±10.3	846.5±137.6

## Data Availability

The data presented in this study will be freely available at https://purl.org/holesond/cvut-ciirc-dvs-benchmark-2021 after the paper is accepted. The event-camera bias configurations used will be included in the dataset package.

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
