# Peer review of "Experimental Comparison between Event and Global Shutter Cameras"

_sensors, 2021, doi:10.3390/s21041137_

Round 1

Reviewer 1 Report

This manuscript presents the comparison results of basic performance and characteristics using four different-type cameras: two types of event-cameras and two types of frame-cameras. The experimentally obtained results can be highly evaluated as vision sensors. Although these cameras used in this study are commercial products, not cameras originally developed by the authors, the evaluation criteria and the comparison results are highly general, comprehensive, and universal as a benchmark for event-cameras. Nevertheless, the five questions below remain as subjects that must be addressed in this manuscript.

1. Section 2 comprehensively introduces a wide range of conventional studies cited from the references of conference proceedings and journal papers including surveys. However, it is unclear what problems the authors are focusing on conventional studies. Please clearly describe the points of view and their solutions.

2. Are Inivation DVS240 and Prophesee ATIS HVGA Gen3 really state-of-the-art event-cameras? In the following literature, Table 1 lists 13 types of up-to-date event-cameras. In terms of models, dates, and specifications, superior performance event-cameras have been released to the market.
- G. Gallego et al., "Event-based Vision: A Survey," in IEEE Transactions on Pattern Analysis and Machine Intelligence, doi: 10.1109/TPAMI.2020.3008413.

3. There are numerous applications of event-cameras in terms of object tracking, surveillance and monitoring, and object or gesture recognition, 3D scanning, optical flow estimation, HDR image reconstruction, wearable electronics and computers, locomotion of robots, and SLAM. Why did you merely limit two applications: marker recognition on a rotating disk and fast object tracking of flying bullets?

4. Which experimental results led to the following expected description?
- A frame-camera may triumph in cluttered scenes (L12-13).
- On the other hand, we expect a frame-camera to triumph in cluttered scenes (L531).

5. Have you gotten the answers to the two questions: “What is the application domain, in which an event-camera surpasses a fast frame-camera?” or “Which scene conditions can be detrimental to the event-camera performance?” (L33-35)?

Author Response

Dear Madam or Sir,

We are writing you regarding your review of our paper "Experimental Comparison Between Event- and Global Shutter-Cameras" that we have recently sent to the journal Sensors MDPI. We thank you for your valuable feedback. We hope that our paper moved towards your expectations after we implemented your observations.

Below we would like to mention our answers to your questions and remarks.

Point 1: Section 2 comprehensively introduces a wide range of conventional studies cited from the references of conference proceedings and journal papers, including surveys. However, it is unclear what problems the authors are focusing on conventional studies. Please clearly describe the points of view and their solutions.

Response 1: We focused on finding what methodologies were used to quantify the performance of event-cameras and to compare event-cameras and frame-cameras. We present the simple and versatile approach in our paper because we did not find any universal and comprehensive methodology of testing and comparing event- and frame-cameras in the literature.
Further, we focused on finding practical high-speed applications of event-cameras. We found that in literature, there are many types of applications mentioned together with advantages of event-cameras over frame-cameras. Nevertheless, our experience, based on tracking the fast-moving objects, differs from these general proposals in certain aspects. Our solution, presented in this paper, was to design two verification experiments (one based on sensing the marker/dot placed on a rotating disk, second based on tracking the flying bullet) to verify event-camera's capabilities and to determine its limits.
Inspired by your question, we reordered the paragraphs of the Related work section into groups corresponding to the three focus areas. We also tried to more clearly explain the focus of the cited studies and how they differ from our study.

Point 2: Are Inivation DVS240 and Prophesee ATIS HVGA Gen3 really state-of-the-art event-cameras? In the following literature, Table 1 lists 13 types of up-to-date event-cameras. In terms of models, dates, and specifications, superior performance event-cameras have been released to the market.
- G. Gallego et al., "Event-based Vision: A Survey," in IEEE Transactions on Pattern Analysis and Machine Intelligence, DOI: 10.1109/TPAMI.2020.3008413.

Response 2: Thank you for pointing this out; it is true that we did not explain why other event-cameras were not used in our analysis. Based on your question, we added the following explanation to the section 3.2 Materials of the paper:
Table 1 in the event-camera survey compares several commercial or prototype event-cameras. Some of them have better specifications than the two event-cameras we had. However, we constructed our benchmark experiments such that only pixel and readout design affect event-cameras' performance. Larger camera pixel array resolution, for example, would not affect the reported performance metrics. Given our benchmark design, the DVS240 (DAVIS240) and DAVIS346 are still the best sensors produced by the company iniVation mentioned in the table. The Samsung cameras were not commercially available in 2020. The exception we found was the Samsung SmartThings Vision home monitoring device, which has an event-camera embedded inside. However, we did not find a simple way of connecting the embedded camera to a computer and recording the events it emits. Before buying the Prophesee ATIS camera, we briefly experimented with the CelePixel CeleX-IV camera. Although its specifications on paper are impressive, it performed much worse in our initial test than the first iniVation product commercially available, the DVS128 from 2008. Prophesee told us in May 2020 that they planned to release their Gen 4 CD sensor in Q4 2020 or later. Thus we could not test it. These findings make us believe that the Prophesee Gen3 ATIS remains one of the state-of-the-art commercially available event-cameras as of 2020.
We further added this paragraph into section 3.2 Materials:
"The cameras we tested have been widely used by researchers and so are relevant to a large scientific community. Event-camera users may use our benchmark to test newer cameras."
We also added to Related work: "The (survey) work also lists properties of thirteen event-cameras. We used two of them in our experiments and one was the state-of-the-art commercially available camera."
The improved abstract also stresses that the ATIS we used was one of the commercially available state-of-the-art cameras.

Point 3: There are numerous applications of event-cameras in terms of object tracking, surveillance and monitoring, and object or gesture recognition, 3D scanning, optical flow estimation, HDR image reconstruction, wearable electronics and computers, locomotion of robots, and SLAM. Why did you merely limit two applications: marker recognition on a rotating disk and fast object tracking of flying bullets?

Response 3: We assume that cameras that perform well in object detection, recognition, and tracking will perform well also in optical flow, SLAM, gesture recognition, or surveillance. The "crispness" of the image required to track marker or dot accurately will be similar to that needed when performing other computer vision tasks. We chose the object tracking and marker recognition tasks to enable other researchers to benchmark their cameras easily using the methods we propose. The ballistic experiment (tracking of a flying bullet) verified the findings obtained in the rotating disk experiments. Compared to the rotating disk, the ballistic experiment provided the opportunity to test the event-camera performance on quantitatively higher image speeds.
We added these two sentences "To that end, we designed an experiment involving markers on a rotating disk, which is easy to replicate. We suggest that cameras performing well in our experiments will also perform well in other computer vision applications, and that these experiments will be useful when evaluating future event-cameras." to the first paragraph in the Introduction to reflect this justification.

Point 4: Which experimental results led to the following expected description?
- A frame-camera may triumph in cluttered scenes (L12-13).
- On the other hand, we expect a frame-camera to triumph in cluttered scenes (L531).

Response 4: The coding efficiency analysis in section 4.5 led us to the conclusion. It is true that in the original text, we did not explain this connection enough. We corrected the Conclusions to include the necessary context: "the coding efficiency analysis suggests that a frame-camera may outperform event-cameras in cluttered scenes." "Assuming the readout bandwidth is fixed, the event-camera output sampling rate decreases as the scene complexity/clutter increases, whereas the frame-camera output sampling rate remains constant." We removed "A frame-camera may triumph in cluttered scenes." from the abstract because it was misleading without stating the context.

Point 5: Have you gotten the answers to the two questions: "What is the application domain, in which an event-camera surpasses a fast frame-camera?" or "Which scene conditions can be detrimental to the event-camera performance?" (L33-35)?

Response 5: Indeed, we did not specifically answer the questions in the text. Yes, we did find the answers, and based on your question, we added the answers to the Conclusions section. Regarding the question "What is the application domain, in which an event-camera surpasses a fast frame-camera?" we can say that the event-camera surpasses the frame-camera in scenes with significant changes restricted to less than 30% of the field of view within the sampling period of interest. As the event pixel latency is significantly lower for negative than for positive contrast changes, the fastest scene changes should be ideally restricted to the negative contrast.
We also improved the formulation of this result in the Conclusions.
Regarding the question "Which scene conditions can be detrimental to the event-camera performance?" we can claim that highly cluttered scenes or scenes with sharp and strong positive contrast edges can be detrimental to the event-camera performance.

In the end, I would like to say thank you for your suggestions and comments, which we found to be very helpful and inspiring. I hope that the changes we made in our article led to its improvement and that we answered all your questions.

Yours faithfully,
Ondrej Holesovsky and coauthors

Reviewer 2 Report

In this draft, the authors compare event cameras with high speed cameras, and perform several experiments to compare. The authors analyze the experimental results.

My most concern about this draft is the novelty. The manuscript is reading more like a test engineering report, not a scientific paper. It seems that there is not scientific or theoretical novelty in this draft, which may not qualify this paper for this journal.

Author Response

Dear Madam or Sir,

We are writing you regarding your review of our paper "Experimental Comparison Between Event- and Global Shutter-Cameras" that we have recently sent to the journal Sensors MDPI.
Below I would like to mention our answers to your questions and remarks.

Point 1: My most concern about this draft is the novelty. The manuscript is reading more like a test engineering report, not a scientific paper. It seems that there is not scientific or theoretical novelty in this draft, which may not qualify this paper for this journal.

Response 1: We agree that the scientific contribution was not clearly stated in our paper. Our goal was to analyze the event-cameras' performance because they are now widely used in many papers with the assumption of being inherently superior to frame-cameras. However, this depends on the conditions and the specificity of the task. Additionally, in some conditions, the event-cameras arguably don't perform as one might expect based on their specifications (e.g., the visual information lost due to readout aggregation or the trailing events generated at higher speeds). We wanted to provide the research community with both the analysis of some of the existing cameras and the easily reproducible methodology to test new event cameras' behavior in the desired conditions.
In the analysis process, we also state several novel findings that might have some value for the research community.

To address your remark, we added statements specifically mentioning our findings to the last paragraph of Related work and made it more apparent that we view our analysis and the testing methodology as contributions.

We would like to thank you for your review and hope that the explanation and the changes we made in our article led to our work's improvement.

Yours faithfully,
Ondrej Holesovsky and coauthors

Reviewer 3 Report

This paper presents the comparison results between event cameras and fast frame-cameras. Performances are tested in two scenarios. One is to observe a marker on a spinning disk and the other one is to observe a flying bullet. The evaluated results include detection rates, position estimation errors, and the minimum pixel latency. Additional meaningful conclusions are summarized.

This manuscript is well written and organized. The authors present a brief background of the field where their research is relevant. The authors present the problem with challenges they focus on. Helpful figures were included in places where it was helpful for the reader's understanding. The testing approach is clearly introduced and the thorough testing results make sense and are encouraged.  

Author Response

Dear Madam or Sir,

I am writing to you regarding your review of our paper "Experimental Comparison Between Event- and Global Shutter-Cameras" that we have recently sent to the journal Sensors MDPI.
Below I would like to mention our answers to your questions and remarks.

Point 1: Extensive editing of English language and style required.

Response 1: Following your recommendation, we asked a native English speaker to edit our paper's language and style. We like the paper much better after he made the corrections.

We would like to thank you for your review and hope that the changes we made improved the manuscript.

Yours faithfully,
Ondrej Holesovsky and coauthors

Reviewer 4 Report

It appears that that this article is an extension of a previous publication titled: "Practical high-speed motion sensing: event cameras vs. global shutter". The goal of this paper is to compare and contrast the event and global shutter cameras, and test in two practical settings. The authors have conducted a thorough experimental comparison of the two types of cameras, including a quantitative and qualitative comparison. 

It was unclear to me how this comparison would benefit researchers who use this cameras. It would be helpful of the authors could expand discussion of this topic. I also found that the related work did not clearly articulate other previous comparisons that were done in this area, and how the manuscript is different from them. Finally, I would also suggest to include a visualization of the two testing setups and a brief overview figure of the two types of cameras.

Author Response

Dear Madam or Sir,

We are writing you regarding your review of our paper "Experimental Comparison Between Event- and Global Shutter-Cameras" that we have recently sent to the journal Sensors MDPI.
Below I would like to mention our answers to your questions and remarks.

Point 1: It was unclear to me how this comparison would benefit researchers who use these cameras. It would be helpful of the authors could expand discussion of this topic.

Response 1: Our primary motivation for this paper was our own experience with event-cameras. We wanted to apply them to several tasks, but the performance was not as expected. Thus we analyzed in-depth what was happening. We think that other researchers attempting to use event-cameras might also encounter similar problems. Our analysis of existing cameras or methodology to test new ones might be helpful for them. Following your suggestions, we tried to describe our motivation (in the first paragraph of Introduction) and the expected impact of the paper (in the last paragraph of Related work) more explicitly.

Point 2: I also found that the related work did not clearly articulate other previous comparisons that were done in this area, and how the manuscript is different from them.

Response 2: Thank you for pointing out this shortcoming. To address it, we reorganized the related work section. The comparison papers are in a single block and we added more detailed descriptions of the differences between the cited papers and our work.

Point 3: Finally, I would also suggest to include a visualization of the two testing setups and a brief overview figure of the two types of cameras.

Response 3: Following this suggestion, we added a photograph of the ballistic experiment with labels and a detailed schematic representation of the rotating disk experiment to improve the testing setups' visualization.

In the end, we would like to say thank you for your suggestions and comments, which we found to be very helpful and inspiring. I hope that the changes we made led to the improvement of the article and that we answered all your questions.

Yours faithfully,
Ondrej Holesovsky and coauthors

Round 2

Reviewer 2 Report

Thank authors for addressing comments.